# Effects of surgical and FFP2 masks on cardiopulmonary exercise capacity in patients with heart failure

**Alexander Kogel**[1]*, **Pierre Hepp**[2], **Tina Stegmann**[1], **Adrienn Tünnemann-Tarr**[1], **Roberto Falz**[3], **Patrick Fischer**[4], **Felix Mahfoud**[4], **Ulrich Laufs**[1], **Sven Fikenzer**[1]

**1** Klinik und Poliklinik für Kardiologie, Universitätsklinikum Leipzig, Leipzig, Germany, **2** Klinik und Poliklinik für Orthopädie, Unfallchirurgie und Plastische Chirurgie, Universitätsklinikum Leipzig, Leipzig, Germany, **3** Institute of Sport Medicine and Prevention, University of Leipzig, Leipzig, Germany, **4** Klinik für Innere Medizin III, Universitätsklinikum des Saarlandes, Saarland University, Homburg (Saar), Germany

* alexander.kogel@medizin.uni-leipzig.de

**Data Availability Statement:** All relevant data are within the paper and its Supporting Information files. Additional raw data is available from the corresponding author.

## Abstract

### Aims

Surgical and FFP2 masks are recommended to reduce transmission of SARS-CoV-2. The cardiopulmonary effects of facemasks in patients with chronic heart failure are unknown. This prospective, cross-over study quantified the effects of wearing no mask (nm), surgical mask (sm), and FFP2 mask (ffpm) in patients with stable heart failure.

### Methods

12 patients with clinically stable chronic heart failure (HF) (age 63.8±12 years, left ventricular ejection fraction (LVEF) 43.8±11%, NTProBNP 573±567 pg/ml) underwent spiroergometry with and without masks in a randomized sequence. Comfort/discomfort was assessed using a standardized questionnaire.

### Results

Maximum power was reduced with both types of masks (nm: 108.3 W vs. sm: 101.2 W vs. ffpm: 95.6 W, p<0.01). Maximum respiratory oxygen uptake (1499ml/min vs. 1481 ml/min vs. 1300 ml/min, p = 0.95 and <0.01), peak ventilation (62.1 l/min vs. 56.4 l/min vs. 50.3 l/min, p = 0.15 and p<0.05) and O2-pulse (11.6 ml/beat vs. 11.8 ml/beat vs. 10.6 ml/beat, p = 0.87 and p<0.01) were significantly changed with ffpm but not sm. Discomfort was moderately but significantly increased (nm: 1.6 vs. sm: 3.4 vs. ffpm: 4.4, p<0.05).

### Conclusion

Both surgical and FFP masks reduce exercise capacity in heart failure patients, while FFP2 masks reduce oxygen uptake and peak ventilation. This reduction in cardiopulmonary performance should be considered in heart failure patients whose daily life activities are often just as challenging as exercise is for healthy adults.

**Funding:** This work was supported by Leipzig University. We acknowledge support from Leipzig University for Open Access Publishing.

**Competing interests:** The authors have declared that no competing interests exist.

## Introduction

During the ongoing COVID-19 pandemic, face masks as protective measures proved effective in decreasing the transmission of SARS-CoV-2 [1, 2]. The protection provided by FFP2 face masks is superior to surgical face masks [3]. However, concerns were raised about the ability to communicate while wearing a face mask and the impact on elderly and frail patients [4]. While data for healthy adults and the general impact of face masks are available, evidence for patients with chronic diseases and cardiovascular diseases is sparse [5–9].

Healthy adults rarely reach high levels of activity during daily activities that would incite exertion [10]. In contrast, patients with chronic heart failure (HF) reach maximum load and exertion more often in their daily lives, and they frequently adapt by decreasing intensity and prolonging the effort [11]. Hence, it is important to assess the potential effects of different face masks in these patients.

This study aims to quantitate the effects of wearing no mask (nm), surgical mask (sm), and FFP2 mask (ffpm) in clinically stable patients with chronic heart failure (HF) on optimal medical therapy. We measured well-established parameters of myocardial and pulmonary function by spiroergometry [12–15].

## Methods

### Subjects

Twelve male patients with chronic HF treated at the outpatient clinic at Leipzig University hospital participated in the study. In this study, patients with a documented diagnosis of HF with reduced or preserved ejection fraction and at least one episode of cardiac decompensation required hospitalization prior randomization were included. All patients were in a compensated status and on pharmacological therapy according to the guidelines for the medical treatment of chronic HF. The study was conducted in accordance with the latest revision of the Declaration of Helsinki. It was approved by the Ethical Committee of the Medical Faculty, University of Leipzig (reference number 328/20-ek). Written informed consent was obtained from all participants.

### Inclusion and exclusion criteria

#### Inclusion.

- Clinically stable chronic heart failure

- Heart Failure with reduced Ejection Fraction (HFrEF), mildly reduced Ejection Fraction (HFmrEF), and Heart Failure with preserved Ejection Fraction, HFpEF

#### Exclusion.

- Contraindications to ergometry

- Acute coronary syndrome

- Symptomatic high-grade valvular ventricular disease

- Decompensated heart failure

- Acute pulmonary embolism

- Acute inflammatory heart disease

- Acute aortic dissection

- Blood pressure at rest >180/100 mmHg

- Acute leg vein thrombosis

- Acute severe general illness

- Extracardiac disease with significantly limited life expectancy (≤6 months)

- Untreated severe ventricular arrhythmias

- Symptomatic bradycardia, AV block II˚ type 2 Mobitz, or AV block III˚ without pacemaker care

- Limited mobility with the need for walkers, wheelchair, or motorized devices without the ability to perform ergometry

- Implanted pacemaker or CRT systems (ICD allowed)

- COPD stage III

## Study design

Medical history was taken using a questionnaire. Subjects received a physical examination and documentation of vital parameters, body measurements, and a resting electrocardiogram (ECG). Each subject performed three incremental exertion tests (IET), one "no-mask" (nm), one with surgical mask (sm), and one with FFP2 mask (ffpm). The order of the testing was randomly assigned using the GraphPad Quickcalcs online randomization tool [16]. Tests were performed at the same time of day with a minimum of 48 hours between two tests. To assess baseline respiratory function, spirometry for each setting (nm, sm, ffpm) was performed. The participants were blinded about their individual test results to avoid influence by an anticipation bias.

## Incremental cardiopulmonary exertion test (CPET)

CPET were performed on a semi-recumbent ergometer (GE eBike, GE Healthcare GmbH, Solingen, Germany, Germany) at a constant speed of 55–65 revolutions per minute (rpm). Each test started with a workload of 20 W with an increase of 8 W within 1 minute (as a ramp) until voluntary exhaustion occurred. Each subject continued for an additional 5-min recovery period at a workload of 25 W.

## Masks

We used typical and widely used disposable FFP2 protective face masks (GuardweFFP2NR, Wuhan Zonsen Medial Products Co., Ltd., Wuhan City, China) and surgical masks (Suavel® Protec Plus, Meditrade, Kiefersfelden, Germany), both with ear loops. The spirometry mask was placed over the masks and fixed with head straps in a leak-proof manner as described earlier [5]. Before every run, we tested for leakage.

## Measurements

Heart rate (HR) (GE-Cardiosoft, GE Healthcare GmbH, Solingen, Germany), maximum oxygen consumption ($VO_{2max}$) and minute ventilation (VE) were monitored continuously at rest, during CPET and recovery. Lung function and spirometry data were collected through a digital spirometer (Vyntus™ CPX, Vyaire Germany, Hoechberg, Germany). For each modality,

(nm, sm, ffpm) data of three expiratory maneuvers with 1-minute intervals were collected using the best values obtained for maximum forced vital capacity (FVC), forced expiratory volume in 1st second (FEV1), peak expiratory flow (PEF) and Tiffeneau index (TIFF). Capillary blood samples (55 μl) were taken from the earlobe at baseline and immediately after cessation of maximum load and analyzed by a common blood gas analyzer (ABL90 FLEX blood gas analyzer, Radiometer GmbH, Krefeld, Germany). Blood pressure (BP) was observed at rest, every 3 minutes during the CPET, and after the first 5 minutes of the recovery period.

### Quantification of comfort/discomfort

We used the questionnaire published by Li et al. to quantify the following ten domains of comfort/discomfort while wearing a mask: humidity, heat, breathing resistance, itchiness, tightness, saltiness, feeling unfit, odor, fatigue, and overall discomfort [17]. The participants were asked 10 minutes after each CPET how they perceived the comfort in the test.

### Statistical analysis

All values are expressed as means and standard deviations unless otherwise stated, and the significance level was defined as $p < 0.05$. Data were analyzed using Microsoft Office Excel® 2010 for Windows (Microsoft Corporation, Redmond, Washington, USA) and GraphPad Prism 9 (GraphPad Software Inc., California, USA). For distribution analysis, the D'Agostino–Pearson normality test was used. For normal distribution, comparisons were made using one-way repeated measures ANOVA with Turkey's post hoc test for multiple comparisons. Otherwise, the Friedman non-parametric test and Dunn's post hoc test were used. Pearsons r was used for correlation analyses and $R^2$ as the coefficient of determination. The study was powered to detect a difference of 10% in VO2max/kg between nm and ffpm with $\beta = 0.2$ and $\alpha = 0.05$.

## Results

12 patients with clinically stable chronic heart failure aged 63.8±12 years, a mean left ventricular ejection fraction (LVEF) 43.8±11% and a mean NTProBNP 573±567 pg/ml were analysed. Additional patients' baseline characteristics are depicted in **Table 1**.

### Spirometry

To test for effects on pulmonary function at rest we performed spirometry. The results of resting spirometry are shown in **Table 2.** Forced vital capacity was reduced by 10.2% (p<0.01) with surgical masks and by 17.2% (p<0.01) using FFP2 masks, respectively. Expiration measured as the volume that has been exhaled at the end of the first second of forced expiration was also significantly reduced by 9.3% (p<0.01) and 17.3% (p<0.01), respectively. Additionally, peak flow was slower with a reduction of 14.0% (p<0.01) and 25.1% (p<0.01).

### Incremental cardiopulmonary exercise test

The effect of face masks on cardiopulmonary parameters under increasing loads was determined by incremental cardiopulmonary exercise test. Results and changes in parameters of the incremental cardiopulmonary exercise tests are shown in **Table 2** and **Fig 1**. Under resting conditions, the surgical mask did not affect cardiopulmonary parameters. While wearing a FFP2 mask, the tidal volume (+14.6%, p<0.05) was significantly greater than using no mask, and systolic blood pressure was significantly reduced (-8.5%, p<0.05). All other measured parameters were not significantly changed at rest.

**Table 1. Baseline characteristics.**

| Parameter | Unit | mean ± SD | | |
|---|---|---|---|---|
| Age | Years | 63.8 | ± | 12.6 |
| Body mass index | kg/m$^2$ | 31.3 | ± | 6.1 |
| Heart rate | Bpm | 75.6 | ± | 12.7 |
| Systolic blood pressure | mmHg | 123 | ± | 14.5 |
| Diastolic blood pressure | mmHg | 73 | ± | 7.8 |
| Ejection fraction | % | 43.8 | ± | 11.2 |
| Ischaemic heart disease | No. (%) | 6 | | (50%) |
| Dilated cardiomyopathy | No (%) | 4 | | (33%) |
| Chronic obstructive pulmonary disease | No (%) | 2 | | (17%) |
| Glomerular filtration rate | ml/min/1.73m$^2$ | 65 | ± | 21.6 |
| NT-proBNP | pg/ml | 573 | ± | 567 |
| Beta-blockers | No. (%) | 11 | | (92%) |
| ACE inhibitors | No. (%) | 2 | | (17%) |
| AT1 antagonists | No. (%) | 2 | | (17%) |
| Sacubitril/valsartan | No. (%) | 8 | | (65%) |
| Aldosterone antagonists | No. (%) | 9 | | (75%) |
| Calcium channel blockers | No. (%) | 2 | | (17%) |
| Diuretics | No. (%) | 8 | | (67%) |
| Lipid lowering drugs | No. (%) | 9 | | (75%) |
| Antiplatelet agents | No. (%) | 9 | | (75%) |
| Oral anticoagulation | No. (%) | 1 | | (9%) |
| Anti-diabetic drugs | No. (%) | 3 | | (25%) |

Maximum load was significantly reduced with surgical (-7.3%, p<0.01) as well as with FFP2 masks (-12.7%, p<0.01). Surgical masks had no significant impact on cardiopulmonary parameters under maximum load. However, FFP2 masks impaired key cardiac parameters like heart rate (-4%, p<0.05), ratio between heart rate and achieved load (+10.2%, p<0.01), maximum oxygen uptake (-13.7%, p<0.01), and oxygen pulse (-9.5%, p<0.01). Additionally the systolic blood pressure (-6%, p<0.05) and the rate pressure product (-10%, p<0.05) were significantly reduced. Wearing an FFP2 mask also significantly reduced respiratory minute volume (-19%, p<0.05) and tidal volume (-15%, p<0.05). Diastolic blood pressure, breathing frequency, and metabolic parameters at maximum load did not significantly change while wearing a face mask. The significant differences measured using the FFP2 masks were associated with very high effect sizes (eta-squared) for the main endpoints: Watt ($\eta^2 = 0.62$), VO2max ($\eta^2 = 0.41$), and oxygen pulse ($\eta^2 = 0.33$).

Notably, the rate pressure product relative to workload did not differ between the tests.

Pulmonary function was impacted using FFP2 masks. This resulted in a significant reduction in respiratory minute volume (-14.8%, p<0.05) and tidal volume (-13.1%, p<0.01).

## Overall discomfort

Patients reported significantly higher overall discomfort wearing masks than without. Surgical masks (+1.8 p<0.05) were reported as more comfortable than FFP2 masks (+2.9 p<0.01).

## Discussion

The main result of this study is that the physical performance of heart failure patients is impaired by face masks, especially FFP2 masks. The greatest limitations were seen in

**Table 2. Results of the incremental cardiopulmonary exercise test.**

| INCREMENTAL EXERTION TEST | | Unit | nm | sm | ffpm | nm vs. sm | nm vs. ffpm |
|---|---|---|---|---|---|---|---|
| Rest | Spirometry results | | | | | | |
| | FVC | L | 3.8 ± 0.7 | 3.5 ± 0.7 | 3.2 ± 0.7 | <0.01 | <0.01 |
| | FEV1 | L | 2.9 ± 0.5 | 2.6 ± 0.5 | 2.4 ± 0.4 | <0.01 | <0.01 |
| | PEF | l/s | 6.9 ± 1.8 | 5.9 ± 1.4 | 5.1 ± 1.6 | <0.01 | <0.01 |
| | Hemodynamic parameters | | | | | | |
| | HR | Bpm | 75.6 ± 12.7 | 77.3 ± 10.6 | 78.3 ± 13.5 | ns | ns |
| | SBP | mmHg | 123 ± 14.6 | 115 ± 14.4 | 112 ± 18.5 | ns | <0.05 |
| | DBP | mmHg | 73.3 ± 8.7 | 70.8 ± 6.9 | 72.7 ± 9.3 | ns | ns |
| | Pulmonary parameters | | | | | | |
| | VE | l/min | 11.4 ± 1.9 | 11.1 ± 2.5 | 11.9 ± 2.9 | ns | ns |
| | Breathing frequency | Brpm | 16.1 ± 4.9 | 13.1 ± 2.5 | 14.4 ± 3.5 | ns | ns |
| | VT | L | 0.8 ± 0.3 | 0.9 ± 0.2 | 0.9 ± 0.4 | ns | <0.05 |
| | Metabolic parameters | | | | | | |
| | pH | | 7.42 ± 0.02 | 7.42 ± 0.02 | 7.43 ± 0.02 | ns | ns |
| | PCO2 | mmHg | 37.2 ± 3.2 | 37.2 ± 3.7 | 36.2 ± 4.3 | ns | ns |
| | PO2 | mmHg | 72.4 ± 9.4 | 74.1 ± 13.6 | 74.6 ± 9.1 | ns | ns |
| Maximum load | Performance | | | | | | |
| | Pmax | W | 108.3 ± 49.3 | 101.2 ± 51.0 | 95.6 ± 49.5 | <0.01 | <0.01 |
| | Hemodynamic parameters | | | | | | |
| | HR | Bpm | 129.7 ± 20.2 | 125.8 ± 20.3 | 124.1 ± 18.9 | ns | <0.05 |
| | HR/Watt | beats/W | 1.35 ± 0.57 | 1.40 ± 0.54 | 1.48 ± 0.59 | ns | <0.01 |
| | VO2max/kg | (ml/min)/kg | 16.0 ± 7.0 | 15.7 ± 7.7 | 13.9 ± 6.5 | ns | <0.01 |
| | Oxygen pulse | ml/beat | 11.6 ± 3.7 | 11.8 ± 4.4 | 10.6 ± 3.5 | ns | <0.01 |
| | SBP | Mmhg | 176 ± 40.6 | 169 ± 32.2 | 165 ± 37.6 | ns | <0.05 |
| | DBP | mmHg | 79.1 ± 10.3 | 80.1 ± 13.2 | 77.5 ± 16.7 | ns | ns |
| | RPP (/1000) | bpm*mmhg | 23.2 ± 7.8 | 21.5 ± 6.6 | 20.8 ± 7.1 | ns | <0.05 |
| | RPP/Watt | bpm*mmhg/W | 232.7 ± 92.3 | 229.9 ± 72.0 | 238.4 ± 92.8 | ns | ns |
| | Pulmonary parameters | | | | | | |
| | VE | l/min | 62.1 ± 21.0 | 56.4 ± 17.3 | 50.3 ± 13.0 | ns | <0.05 |
| | Breathing frequency | Brpm | 30.2 ± 5.8 | 28.7 ± 5.2 | 28.9 ± 4.1 | ns | ns |
| | VT | L | 2.0 ± 0.5 | 2.0 ± 0.4 | 1.7 ± 0.3 | ns | <0.01 |
| | Metabolic parameters | | | | | | |
| | RER | | 1.08 ± 0.11 | 1.05 ± 0.08 | 1.05 ± 0.08 | ns | ns |
| | pH | | 7.36 ± 0.04 | 7.36 ± 0.05 | 7.36 ± 0.05 | ns | ns |
| | PCO2 | mmHg | 36.5 ± 3.9 | 37.3 ± 3.0 | 37.3 ± 5.4 | ns | ns |
| | PO2 | mmHg | 75.5 ± 10.8 | 76.8 ± 8.9 | 76.9 ± 8.8 | ns | ns |
| Overall discomfort | | | 1.6 ± 1.5 | 3.4 ± 1.7 | 4.5 ± 2.6 | <0.05 | <0.01 |

nm:no mask; sm: surgical mask; ffpm: FFP2 mask; FVC: forced vital capacity; FEV1: volume exhaled in the first second of forced expiration; PEF: peak flow; HR: heart rate; SBP: systolic blood pressure; DBP: diastolic blood pressure; VE: respiratory minute volume; VT: tidal volume; Pmax: maximum load achieved; RPP: rate pressure product; RER: respiratory exchange ratio.

# Cardiac

# Pulmonary

**Fig 1. Wearing a FFP2 mask significantly changes key cardiopulmonary parameters in heart insufficient patients.**
VO2max: maximum respiratory oxygen uotake; HR: heart rate; RER: respiratory exchange ratio; sm: surgical mask;
ffpm: FFP2 mask.

maximum respiratory oxygen uptake and respiratory volume. Nonetheless, wearing face
masks does not severely lower the overall comfort of these patients while physically exercising.

## Exertion

Under all three conditions, patients reached exertion during CPET as seen by similar RER values [18]. The significantly lower heart rate while wearing a FFP2 mask could be interpreted as
a sign of lower exertion levels. Notably, the ratio between heart rate and achieved load is significantly higher with FFP2 masks, an effect that would usually be expected with worsening heart
failure [19]. The lower maximal heart rate is most likely an indication of a reduced ability of
the failing heart to adapt. In this setup, medication with drugs reducing the heart rate is of no
concern because the medication did not change between the tests, although generally beta
blockade can significantly alter cardiopulmonary parameters during exercise [20]. There were
no significant changes in analysed parameters between the first and the following tests, so that
adequate recovery between tests can be assumed.

## Cardiac function

The systolic blood pressure decreased in the heart failure patients wearing a FFP2 mask under
resting conditions. This is in contrast to findings in healthy adults [5, 21, 22]. This effect persisted through higher loads and may be a result of the heart not being able to adapt to the cardiopulmonary changes of wearing a mask. This interpretation is consistent with the significant
reduction of respiratory oxygen uptake, oxygen pulse, and rate pressure product under maximum load [19]. While beta blockers can impair the increase in heart rate during exercise, they
have no effect on maximum oxygen uptake [23]. Additionally, there were no changes of medication between the tests. Therefore, the changes between the tests cannot be attributed to the
medication.

Comparable energy expenditure for the same load is required (RPP per Watt). Increased
breathing resistance may lead to prolonged inspiration times and therefore to longer phases of
higher negative intrathoracic pressure. This hypothesis is supported by the findings on inspiration times in healthy adults, which were higher while wearing a ffpm [5]. The increased cardiac
preload challenges the failing heart because its limited ability to increase the stroke volume. In
addition, increased transmural left ventricular pressure due to the negative intrathoracic pressure may further reduce the stroke volume [24]. The effects of breathing resistance on the cardiopulmonary system are highlighted by the use of training with breathing resistance to
maximize endurance capacity and respiratory muscle function [25].

Wearing a FFP2 mask compared with no mask re-classified the heart failure patients in our
study from Weber B (VO2max/kg > 16 (ml/min)/kg) to Weber C (VO2max/kg > 10–16 (ml/
min)/kg) [14].

## Pulmonary function

Similar to healthy adults, the heart failure patients showed significant reductions in spirometry
results while wearing a mask [5]. This effect was observed under resting and exercise conditions. Interestingly, the tidal volume was significantly increased while wearing FFP2 masks.
This may be due to an anticipation of higher breathing resistance and subsequent adaptation

[21]. The tidal volume and the respiratory minute volume were significantly reduced under maximum load. The higher workload for respiratory muscles compared to exercise with no mask is likely leading to exertion of these muscles and consecutively to diminished maximum pulmonary function. This again can be seen in the training effect achieved by willingly using breathing resistance to achieve higher respiratory muscle function [25]. In healthy adults, a myocardial compensation for the pulmonary limitation due to wearing a mask has been discussed [5]. Due to prolonged inspiration times, the stroke volume of the heart is increased [26]. Our data suggest that in patients with impaired myocardial function this compensation may not be possible.

## Discomfort

The participants reported a significant increase of overall discomfort from no mask over surgical masks to FFP2 masks. However, even the most discomfortable FFP2 masks were described as moderately discomfortable (4.5 of 10). Contrary, healthy adults described higher overall discomfort (7.0 of 10) under the same condition [5]. This could either be due to a habituation effect over the course of the pandemic or because heart failure patients are adapted to being restricted in their performance. Therefore, additional restrictions with masks may not cause the same level of discomfort reported by healthy persons.

## Limitations of the study

Limitations of the study include the relatively small sample size, but the study was sufficiently powered to detect a difference of 10% in VO2max/kg between nm and ffpm with $\beta = 0.2$ and $\alpha = 0.05$. The study was randomized but not blinded. The external validity concerning the impact of the masks may be reduced by the laboratory conditions of wearing a spirometry mask above the tested surgical or FFP2 mask. Additionally, only one type of FFP2 mask was used. There are possible differences to the facemasks of other manufacturers. Concerns were raised about leakage during spiroergometry while wearing a face mask [27]. We tested for leakage before every run and found no indication of leakage.

There were no significant changes in analysed parameters between the first and the following tests.

## Conclusion

Wearing a face mask significantly reduces the cardiopulmonary performance of heart failure patients. Changes in critical cardiac and pulmonary parameters are more pronounced while wearing an FFP2 mask. Notably, the overall discomfort of the patients was only moderate while healthy adults described a much stronger discomfort [5]. This reduction in cardiopulmonary performance should be considered in heart failure patients for which daily life activities are often just as challenging as exercise is for healthy adults. Whether this limitation can be improved by training is not known, but it is a good target for future research.

## Supporting information

**S1 Dataset.**
(XLSX)

## Acknowledgments

The authors would like to thank all colleagues of the cardiologic outpatient clinic that contributed to the care of the patients.

## Author Contributions

**Conceptualization:** Ulrich Laufs, Sven Fikenzer.

**Formal analysis:** Alexander Kogel, Sven Fikenzer.

**Investigation:** Alexander Kogel, Sven Fikenzer.

**Methodology:** Patrick Fischer.

**Resources:** Tina Stegmann, Adrienn Tünnemann-Tarr, Roberto Falz, Patrick Fischer.

**Supervision:** Pierre Hepp, Felix Mahfoud, Ulrich Laufs, Sven Fikenzer.

**Visualization:** Alexander Kogel.

**Writing – original draft:** Alexander Kogel, Sven Fikenzer.

**Writing – review & editing:** Pierre Hepp, Tina Stegmann, Adrienn Tünnemann-Tarr, Roberto Falz, Patrick Fischer, Felix Mahfoud, Ulrich Laufs, Sven Fikenzer.

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
