## [Decision Letter · Decision Letter 0]

21 Mar 2022

PONE-D-21-29862Effects of surgical and FFP2 masks on cardiopulmonary exercise capacity in patients with heart failurePLOS ONE

Dear Dr. Kogel,

Thank you for submitting your manuscript to PLOS ONE. After careful consideration, we feel that it has merit but does not fully meet PLOS ONE’s publication criteria as it currently stands. Therefore, we invite you to submit a revised version of the manuscript that addresses the points raised during the review process.

ACADEMIC EDITOR:The manuscript has undergone a deep review from one reviewer. The authors should address all the queries raised by the reviewer before full consideration for publication

We look forward to receiving your revised manuscript.

Kind regards,

Antonino Salvatore Rubino, M.D., Ph.D.

Academic Editor

PLOS ONE

Journal Requirements:

This work was supported by Universitätsklinikum Leipzig.

The authors received no specific funding for this work.

6.Please review your reference list to ensure that it is complete and correct. If you have cited papers that have been retracted, please include the rationale for doing so in the manuscript text, or remove these references and replace them with relevant current references. Any changes to the reference list should be mentioned in the rebuttal letter that accompanies your revised manuscript. If you need to cite a retracted article, indicate the article’s retracted status in the References list and also include a citation and full reference for the retraction notice.

Reviewers' comments:

Reviewer's Responses to Questions

**Comments to the Author**

1. Is the manuscript technically sound, and do the data support the conclusions?

Reviewer #1: Yes

2. Has the statistical analysis been performed appropriately and rigorously? 

Reviewer #1: Yes

3. Have the authors made all data underlying the findings in their manuscript fully available?

Reviewer #1: Yes

4. Is the manuscript presented in an intelligible fashion and written in standard English?

Reviewer #1: Yes

5. Review Comments to the Author

Reviewer #1: Thank you for your submitted manuscript entitled, “Effects of surgical and FFP2 masks on cardiopulmonary exercise capacity in patients with heart failure. The area of the research is interesting, however it needs a few amendments. Overall, the paper is well-written, well-written statistics.

ABSTRACT

• Clarify the subjects’ level and background

• Could be a relevant conclusion of the present study to find what is important to know.

INTRODUCTION

• The introduction is consistent and easy to follow. Hypotheses are clearly formulated.

• The Authors should clarify the actual heritage of this study. I am concerned about the originality of the present study.

METHOD

• How was sample size determined? (Sampling technique!)

• What about the inclusion and exclusion criteria?

STATISTICAL ANALYSIS

• Please, present methods of data analysis and criterion of results interpretation.

• Please add a power analysis, which takes into account the number of variables

• RESULTS

• Obviously, the authors conducted a variance analysis. Please describe and explain the used test(s) in the statistical section.

• Results description is a little chaotic and insufficient. Please, add some introductions to the description of the results and indicate what and why you did. Each result presented in the tables should be commented on in the text. Without that, readers do not know how to interpret the tables.

• DISCUSSION

• Discussion should be more based on the literature

CONCLUSION

• Why might one want to cite this paper? What is the true impact of the literature?

6. PLOS authors have the option to publish the peer review history of their article (what does this mean?). If published, this will include your full peer review and any attached files.

Reviewer #1: **Yes: **Souhail Hermassi

---

## [Author Response · Author response to Decision Letter 0]

1 Apr 2022

Revision of PONE-D-21-29862

Effects of surgical and FFP2 masks on cardiopulmonary exercise capacity in patients with heart failure

Point-by-point response to the reviewers

General comment to the editor:

We thank the editors and the reviewer for their time and helpful comments which improved the manuscript. All changes were highlighted in the revised manuscript.

Reviewers' comments to the author:

Thank you for your submitted manuscript entitled, “Effects of surgical and FFP2 masks on cardiopulmonary exercise capacity in patients with heart failure. The area of the research is interesting, however it needs a few amendments. Overall, the paper is well-written, well-written statistics.

Response: 

Thank you for the positive feedback and the recommendations.

ABSTRACT

• Clarify the subjects’ level and background

Response: 

We moved the information from “Aims” to “Methods”

“12 patients with clinically stable chronic heart failure (HF) (age 63.8±12 years, left ventricular ejection fraction (LVEF) 43.8±11 %, NTProBNP 573±567 pg/ml)…”

• Could be a relevant conclusion of the present study to find what is important to know.

Response: 

Thank you, we have extended the Conclusion accordingly.

“Both surgical and FFP masks reduce exercise capacity in heart failure patients, while FFP2 masks reduce oxygen uptake and peak ventilation. This reduction in cardiopulmonary performance should be considered in heart failure patients for which daily life activities are often just as challenging as exercise is for healthy individuals.”

INTRODUCTION

• The introduction is consistent and easy to follow. Hypotheses are clearly formulated.

Response: 

Thank you.

• The Authors should clarify the actual heritage of this study. I am concerned about the originality of the present study.

Response: 

Although there are several recent publications regarding the impact of face masks and exercise, there is currently a lack of data concerning patients with heart failure. Since the topic is of very high daily relevance, we are convinced that the data can be helpful for all who are monitoring/diagnosing heart failure patients in the context of training and exercise und pandemic conditions.

METHOD

• How was sample size determined? (Sampling technique!)

Response: 

The sample size was based on results of our prior study. “The study was powered to detect a difference of 10% in VO2max/kg between nm and ffpm with β=0.2 and α=0.05.” This was the underlying basis for the sample size calculation. 

This is mentioned in the “Statistical analysis”.

• What about the inclusion and exclusion criteria?

Response: 

Thank you very much. We added the following information in the Method section:

Inclusion/Exclusion criteria 

Inclusion:

• Clinically stable chronic heart failure 

• Heart Failure with reduced Ejection Fraction (HFrEF), mildly reduced Ejection Fraction (HFmrEF) and Heart Failure with preserved Ejection Fraction, HFpEF

Exclusion:

• Contraindications to ergometry:

• Acute coronary syndrome

• Symptomatic high-grade valvular ventricular disease

• Decompensated heart failure

• Acute pulmonary embolism

• Acute inflammatory heart disease

• Acute aortic dissection

• Blood pressure at rest >180/100 mmHg

• Acute leg vein thrombosis

• Acute severe general illness

• Extracardiac disease with significantly limited life expectancy (≤6 months)

• Untreated severe ventricular arrhythmias

• Symptomatic bradycardia, AV block II° type 2 Mobitz, or AV block III° without pacemaker care

• Limited mobility with need for walkers, wheelchair, or motorized devices without ability to perform ergometry

• Implanted pacemaker or CRT systems (ICD allowed)

• COPD stage III

STATISTICAL ANALYSIS

• Please, present methods of data analysis and criterion of results interpretation.

Response: 

As written in the Methods section (Statistical analysis): 

“For distribution analysis, the D'Agostino–Pearson normality test was used. For normal distribution, comparisons were made using one-way repeated measures ANOVA with Turkey's post hoc test for multiple comparisons. Otherwise, the Friedman non-parametric test and Dunn's post hoc test were used. Pearsons r was used for correlation analyses and R2 as the coefficient of determination.” The significance level was defined as p < 0.05.

• Please add a power analysis, which takes into account the number of variables

Response: 

Thank you very much. 

Related to the main endpoints (Watt, VO2max, oxygen pulse) we included the effect sizes (eta-squared) in the text as follows:

The significant differences measured using the FFP2 masks were associated with very high effect sizes (eta-squared) for the main endpoints: Watt (�2=0.62), VO2max (�2=0.41), and oxygen pulse (�2=0.33).

RESULTS

• Obviously, the authors conducted a variance analysis. Please describe and explain the used test(s) in the statistical section.

Response:

As written in the Methods section (Statistical analysis): 

“… comparisons were made using one-way repeated measures ANOVA with Turkey's post hoc test for multiple comparisons.” 

• Results description is a little chaotic and insufficient. Please, add some introductions to the description of the results and indicate what and why you did. Each result presented in the tables should be commented on in the text. Without that, readers do not know how to interpret the tables.

Response:

We added short introductions to the result subsections to highlight why we did the respective test. revised the manuscript to present each result in the text as well as in the tables. 

DISCUSSION

• Discussion should be more based on the literature

Response:

We added relevant and current literature to the discussion.

CONCLUSION

• Why might one want to cite this paper? What is the true impact of the literature?

Response:

The manuscript is very important for the scientific community because the data show for the first time the impact of face masks in incremental exercise tests in heart failure patients, which may result in downgrading based on Weber classification. Additionally, this information is important for all caregivers of heart failure patients.

---

## [Decision Letter · Decision Letter 1]

23 May 2022

Effects of surgical and FFP2 masks on cardiopulmonary exercise capacity in patients with heart failure

PONE-D-21-29862R1

Dear Dr. Kogel,

We’re pleased to inform you that your manuscript has been judged scientifically suitable for publication and will be formally accepted for publication once it meets all outstanding technical requirements.

Kind regards,

Antonino Salvatore Rubino, M.D., Ph.D.

Academic Editor

PLOS ONE

Additional Editor Comments (optional):

Reviewers' comments:

Reviewer's Responses to Questions

**Comments to the Author**

1. If the authors have adequately addressed your comments raised in a previous round of review and you feel that this manuscript is now acceptable for publication, you may indicate that here to bypass the “Comments to the Author” section, enter your conflict of interest statement in the “Confidential to Editor” section, and submit your "Accept" recommendation.

Reviewer #1: All comments have been addressed

2. Is the manuscript technically sound, and do the data support the conclusions?

Reviewer #1: Yes

3. Has the statistical analysis been performed appropriately and rigorously? 

Reviewer #1: Yes

4. Have the authors made all data underlying the findings in their manuscript fully available?

Reviewer #1: Yes

5. Is the manuscript presented in an intelligible fashion and written in standard English?

Reviewer #1: Yes

6. Review Comments to the Author

Reviewer #1: Thank you for your effort !

The manuscript is well written after the minor revision and now is suitable for publication

7. PLOS authors have the option to publish the peer review history of their article (what does this mean?). If published, this will include your full peer review and any attached files.

Reviewer #1: **Yes: **Souhail Hermassi

---

## [Editor Report · Acceptance letter]

12 Aug 2022

PONE-D-21-29862R1 

Effects of surgical and FFP2 masks on cardiopulmonary exercise capacity in patients with heart failure 

Dear Dr. Kogel:

I'm pleased to inform you that your manuscript has been deemed suitable for publication in PLOS ONE. Congratulations! Your manuscript is now with our production department. 

Kind regards, 

on behalf of

Dr. Antonino Salvatore Rubino 

Academic Editor

PLOS ONE